# Expert Race: A Flexible Routing Strategy for Scaling Diffusion Transformer with Mixture of Experts

Yike Yuan [* 1]  Ziyu Wang [* 1 2]  Zihao Huang [1]  Defa Zhu [1]  Xun Zhou [1]  Jingyi Yu [† 2]  Qiyang Min [† 1]

## Abstract

Diffusion models have emerged as mainstream framework in visual generation. Building upon this success, the integration of Mixture of Experts (MoE) methods has shown promise in enhancing model scalability and performance. In this paper, we introduce Race-DiT, a novel MoE model for diffusion transformers with a flexible routing strategy, Expert Race. By allowing tokens and experts to compete together and select the top candidates, the model learns to dynamically assign experts to critical tokens. Additionally, we propose per-layer regularization to address challenges in shallow layer learning, and router similarity loss to prevent mode collapse, ensuring better expert utilization. Extensive experiments on ImageNet validate the effectiveness of our approach, showcasing significant performance gains while promising scaling properties.

## 1. Introduction

Recent years have seen diffusion models earning considerable recognition within the realm of visual generation. They have exhibited outstanding performance in multiple facets such as image generation (Ramesh et al., 2022; Nichol et al., 2021; Saharia et al., 2022; Rombach et al., 2022b; Esser et al., 2024), video generation (Gupta et al., 2024; Brooks et al., 2024), and 3D generation (Zhang et al., 2024; Bensadoun et al., 2024). Thus, diffusion models have solidified their position as a pivotal milestone in the field of visual generation studies. Mimicking the triumph of transformer-based large language models (LLMs), diffusion models have effectively transitioned from U-Net to DiT and its variants. This transition yielded not only comparable scaling proper-

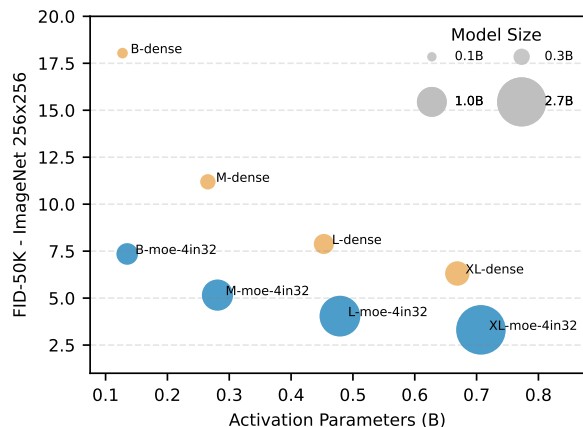

*Figure 1.* Comparison between Dense and our MoE models. Our model significantly outperforms the Dense model with fewer activation parameters.

ties but also an equally successful pursuit of larger models.

In the quest for larger models, the Mixture of Experts (MoE) approach, proven effective in scaling large language models (LLMs) (Jiang et al., 2024), exhibits promising potential when incorporated into diffusion models. Essentially, MoE utilizes a routing module to assign tokens among experts (typically, a set of Feed-Forward Networks (FFN)) based on respective scores. This router module, pivotal to MoE's functionality, employs common strategies such as token-choice and expert-choice.

Meanwhile, we observe that the visual signals processed by diffusion models exhibit two distinct characteristics compared to those in LLMs. First, **visual information tends to have high spatial redundancy.** For instance, significant disparity in information density exists between the background and foreground regions, with the latter typically containing more critical details. Second, **denoising task complexity exhibits temporal variation across different timesteps** (Go et al., 2023). Predicting noise at the beginning of the denoising process is substantially simpler than predicting noise towards the end, as later stages require finer detail reconstruction. These unique characteristics necessitate specialized routing strategies for diffusion models.

Consider these characteristics under the MoE, the presence

*Equal contribution [1]Seed-LLM-Model Team, ByteDance [2]ShanghaiTech University. Correspondence to: Qiyang Min <minqiyang@bytedance.com>, Jingyi Yu <yu-jingyi@shanghaitech.edu.cn>.

*Proceedings of the 42ⁿᵈ International Conference on Machine Learning*, Vancouver, Canada. PMLR 267, 2025. Copyright 2025 by the author(s).

of a routing module can adaptively allocate computational resources. By **assigning more experts to challenging tokens** and fewer to simpler ones, we can enhance model utilization efficiency. Previous strategies like expert-choice anticipated this, but their routing design limit the assignment flexibility to image spatial regions without considering temporal denoising timestep complexity.

In this paper, we introduce Race-DiT, a novel family of MoE models equipped with enhanced routing strategies, Expert Race. We find that simply increasing strategy flexibility greatly boost the model's performance. Specifically, we conduct a "race" among tokens from different samples, timesteps, and experts, and select the top-k tokens from all. This method effectively filters redundant tokens and optimizes computational resource deployment by the MoE.

Expert Race introduces a high degree of flexibility in token allocation within the MoE framework. However, there are several challenges when extending DiT to larger parameter scales using MoE. First, we observe that routing in the shallow layers of MoE struggles to learn the assignment, especially with high-noise inputs. We believe this is due to the weakening of the shallow components in the identity branch of the DiT framework. To address this, we propose an auxiliary loss with layer-wise regularization to aid learning. Second, given the substantial expansion of the candidate space, to prevent the collapse of the allocation strategy, we extend the commonly used balance loss from single experts to expert combinations. This extension is complemented by router similarity loss, which ensures effective expert utilization by regulating pairwise expert selection patterns.

To validate the proposed method, we conducted experiments on ImageNet (Deng et al., 2009), performing detailed ablations on the proposed modules and investigating the scaling behaviors of multiple factors. Results show that our approach achieves significant improvements across multiple metrics compared to baseline methods.

In summary, our main contributions include

- Expert Race, a novel MoE routing strategy for diffusion transformers that supports high routing allocation flexibility in both spatial image regions and temporal denoising steps.

- Router similarity loss, a new objective that optimizes expert collaboration through router logits similarity, effectively maintaining workload equilibrium and diversifying expert combinations without compromising generation fidelity.

- Per-layer Regularization that ensures effective learning in the shallow layers of MoE models.

- Detailed MoE scaling analysis in terms of hidden split and expert expansion provides insights for extending this MoE model to diverse diffusion tasks.

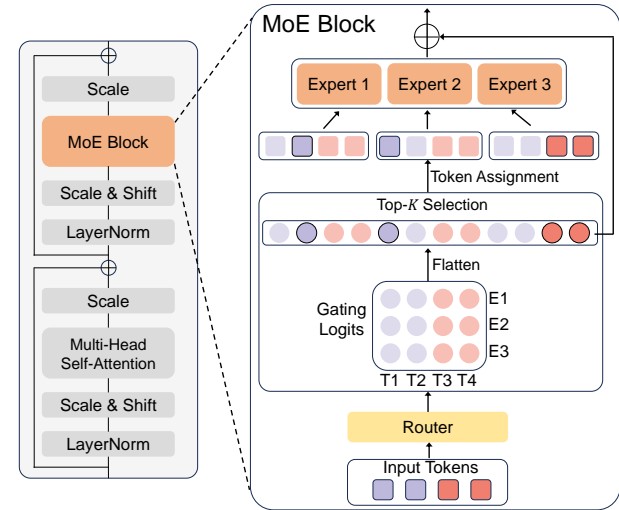

*Figure 2.* The Race-DiT Architecture. We replace the Multi-Layer Perceptron (MLP) with the MoE block, which consists of a Router and multiple Experts. In Race-DiT, the token assignment is done once for all. Each token can be assigned to any number of experts, and each expert can process any number of tokens (including zero).

## 2. Related Work

### 2.1. Mixture of Experts

Mixture of Experts (MoE) improves computational efficiency by activating only a subset of parameters at a time and forcing the other neurons to be zero. Typically, MoE is used to significantly scale up models beyond their current size leveraging the natural sparsity of activations. This technique has been widely applied in LLMs first (Lepikhin et al., 2021; Fedus et al., 2022) and then extended to the vision domain (Riquelme et al., 2021). The most commonly used routing strategy in MoE is token-choice, in which each token selects a subset of experts according to router scores. For its variants, THOR (Zuo et al., 2022) employs a random strategy, BASELayer (Lewis et al., 2021) addresses the linear assignment problem, HASHLayer (Roller et al., 2021) uses a hashing function, and MoNE (Jain et al., 2024) uses greedy top-k. All of these methods allocate fixed number of experts to each token. DYNMoE (Guo et al., 2024) and ReMoE (Wang et al., 2024c) activates different number of experts for each token by replacing TopK with threshold and using additional regularization terms to control the total budget. Also, some auxiliary regularization terms are applied to constrain the model to activate experts uniformly (Zoph et al., 2022; Dai et al., 2024; Wang et al., 2024a). Expert choice (Zhou et al., 2022) has been proposed to avoid load imbalance without additional regularizations and enhance routing dynamics, but due to conflicts with mainstream causal attention, it is less commonly applied in large language models (LLMs).

### 2.2. Multiple Experts in Diffusion

Diffusion follows a multi-task learning framework that shares the same model across different timesteps. Consequently, many studies have explored whether performance can be enhanced by disentangling tasks according to timesteps inside the model. Ernie (Feng et al., 2023) and e-diff (Balaji et al., 2022) manually separate the denoising process into multiple stages and train different models to handle each stage. MEME (Lee et al., 2024) uses heterogeneous models and DTR (Park et al., 2024) heuristically partitions along the channel dimension. DyDiT (Zhao et al., 2024) introduce nested MLPs and channel masks to fit varing complexities across time and spatial dimensions. DiT-MoE (Fei et al., 2024), EC-DiT (Sun et al., 2024), and Raphael (Xue et al., 2024) have applied MoE architectures, learning to assign experts to tokens in an end-to-end manner. Compared with previous works, our methods proposed build on the MoE but further enhancing its flexibility on dynamically allocate experts on all dimensions to unleash its potential.

## 3. Preliminaries

Before introducing our MoE design, we briefly review some preliminaries of diffusion models and mixture of experts.

### 3.1. Diffusion Models

Diffusion models (Ho et al., 2020) are a class of generative models that sample from a noise distribution and learn a gradual denoising process to generate clean data. It can be seen as an interpolation process between the data sample $x_0$ and the noise $\epsilon$. A typical Gaussian diffusion models formulates the forward diffusion process as

$$x_t = \sqrt{\bar{\alpha}_t} x_0 + \sqrt{1 - \bar{\alpha}_t} \epsilon \tag{1}$$

where $\epsilon \sim \mathcal{N}(0, \mathbf{I})$ is the Gaussian noise and $\bar{\alpha}_t$ is a monotonically decreasing function from 1 to 0. Diffusion models use the neural networks to estimate the reverse denoising process $p_\theta(x_{t-1}|x_t) = \mathcal{N}(\mu_\theta(x_t), \Sigma_\theta(x_t))$. They are trained by minimizing the following objectives:

$$\min_\theta \mathbb{E}_{x_0,t,\epsilon} \left[ \|\mathbf{y} - F_\theta(x_t; c, t)\|^2 \right], \tag{2}$$

where $t$ is the timestep which uniformly distributed between 0 to $T$, $c$ is the condition information. e.g. class labels, image or text-prompt. The training target $\mathbf{y}$ can be a Gaussian noise $\epsilon$, the original data sample $x_0$ or velocity $v = \sqrt{1 - \bar{\alpha}} \epsilon - \sqrt{\bar{\alpha}} x_0$.

Early diffusion models used U-net (Dhariwal & Nichol, 2021; Rombach et al., 2022a) as their backbone. Recently, Transformer-based diffusion models (Peebles & Xie, 2023) with adaptive layer normalization (AdaLN) (Perez et al., 2018) have become mainstream, showing significant advantages in scaling up.

### 3.2. Mixture of Experts

Mixture-of-Experts (MoE) is a neural network layer comprising a router $\mathcal{R}$ and a set $\{E_i\}$ of $N_E$ experts, each specializing in a subset of the input space and implemented as FFN. The router maps the input $X \in \mathbb{R}^{B \times L \times D}$ into token-expert affinity scores $\mathbf{S} \in \mathbb{R}^{B \times L \times E}$, trailed by a gating function $\mathcal{G}$:

$$\mathbf{S} = \mathcal{G}(\mathcal{R}(x)). \tag{3}$$

The input will be assigned to a subset of experts with top-k highest scores for computation and its output is the weighted sum of these experts' output. A unified expression is as follows:

$$\mathbf{G} = \begin{cases} \mathbf{S}, & \text{if } \mathbf{S} \in \texttt{TopK}(\mathbf{S}, \mathcal{K}) \\ 0, & \text{Otherwise} \end{cases} \tag{4}$$

$$\text{MoE}(X) = \sum_{i \in N_E} G_i(X) * E_i(X) \tag{5}$$

where $\mathbf{G} \in \mathbb{R}^{B \times L \times E}$ is the final gating tensor and $\texttt{TopK}(\cdot, \mathcal{K})$ is an operation that builds a set with $\mathcal{K}$ largest value in tensor.

To maintain a constant number of activated parameters while increasing the top-k expert selection, the MoE model often splits the inner hidden dimension of each expert based on the top-k value, named fine-grained expert segmentation (Dai et al., 2024). In the subsequent discussions, an "$x$-in-$y$" MoE means there are $y$ candidate experts, with the top-$x$ experts activated, and the hidden dimension of expert's intermediate layer will be divided by $x$.

### 3.3. The Rationality of Using MoE in Diffusion Models

Diffusion models possess several distinctive characteristics.

- Multi-task in nature, tasks at different timesteps predicting the target are not identical. Prior works like e-diff (Balaji et al., 2022) validate this dissimilarity.

- Redundancy of image tokens. The information density varies across different regions, leading to unequal difficulties in generation.

Given these traits, MoE presents a suitable architecture for diffusion models. Its routing module can flexibly allocate and combine tokens and experts based on the predicted difficulties. We consider the allocation process as a distribution of computational resources. More challenging timesteps and complex image patches should be allocated to more experts. Achieving this requires a routing strategy with sufficient flexibility to distribute resources with broader degrees of freedom. Our method is designed following this principle.

# 4. Taming Diffusion Models with Expert Race

## 4.1. General Routing Formulation

For computational tractability, we decompose the original score tensor $\mathbf{S}$ into two operational dimensions through permutation and reshaping, obtaining matrix $\mathbf{S}' \in \mathbb{R}^{D_A \times D_B}$, where

- $D_B$: Size of the expert candidate pool;
- $D_A$: Number of parallel selection operations.

This dimensional reorganization enables independent top-$k$ selection within each row while preserving cross-row independence.

Following the sparse gating paradigm in (Zhou et al., 2022; Lepikhin et al., 2021), we control the MoE layer sparsity through parameter $k$, which specifies the expected number of activated experts per token. To satisfy system capacity constraints, the effective selection size per candidate pool is defined as:

$$\mathcal{K} = \frac{k}{N_E} \cdot D_B. \tag{6}$$

The routing objective, aligning with the optimization framework in (Lewis et al., 2021), formalizes as the maximization of aggregated gating scores:

$$\max \sum_{i=1}^{D_A} \sum_{j \in \mathcal{T}_i} \mathbf{S}'_{i,j}, \tag{7}$$

where $\mathcal{T}_i$ denotes the set of indices corresponding to the top-$\mathcal{K}$ values in the $i$-th row of $\mathbf{S}'$.

**Suboptimal in Conventional Strategies** As shown in Table 1, Figure 3, the unified framework generalizes existing routing methods through top-$\mathcal{K}$ selection in $\mathbf{S}'$. However, standard row-wise approaches like Token-Choice and Expert-Choice exhibit inherent sub-optimality. These selection methods struggle to achieve optimal allocation in practice, as the required uniform distribution of top $\mathcal{K} \times D_A$ elements across rows, which is necessary for attaining the theoretical optimum in Equation (7), rarely holds with real-world data distributions.

In practical scenarios like image diffusion model training, generation complexity varies across two key dimensions: denoising timesteps ($B$) and spatial image regions ($L$). To address this computational heterogeneity, the routing module must dynamically allocate more experts to tokens with greater generation demands. However, the token-choice strategy, since $D_A$ is constituted by dimensions $B\&L$, both dimensions will receive an identical amount of activation experts. Expert-Choice mitigates this issue but remains constrained by its $L$-dimensional top-$\mathcal{K}$ selection, limiting optimal allocation potential.

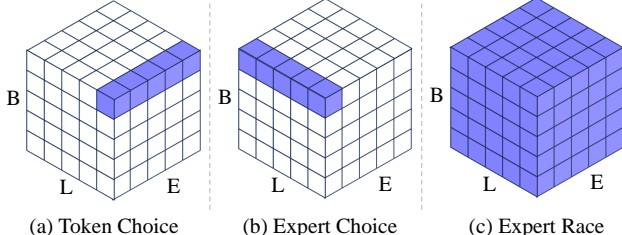

| | | |
|---|---|---|
| (a) Token Choice | (b) Expert Choice | (c) Expert Race |

*Figure 3.* Top-$\mathcal{K}$ Selection Flexibility. $B$: batch size; $L$: sequence length; $E$: the number of experts. (a) Token Choice selects top-$\mathcal{K}$ experts along the expert dimension for each token. (b) Expert Choice selects top-$\mathcal{K}$ tokens along the sequence dimension for each expert. (c) Expert Race selects top-$\mathcal{K}$ across the entire set.

*Table 1.* Specific design choices of different routing strategies.

| Method | $D_A$ | $D_B$ | $\mathcal{K}$ |
|---|---|---|---|
| Token-choice | $B * L$ | $E$ | $k$ |
| Expert-choice | $B * E$ | $L$ | $k * L / E$ |
| Expert-Race | $1$ | $B * L * E$ | $B * L * k$ |

## 4.2. Expert Race

To address these limitations, we propose Expert-Race, which performs global top-$\mathcal{K}$ selection across all gating scores in a single routing pass. The "Race" mechanism provides an optimal solution to Equation (7) by setting $D_A = 1$, ensuring the selected $\mathcal{K}$ elements are globally maximal. This design maximizes router flexibility to learn adaptive allocation patterns, enabling arbitrary expert-to-token assignments and dynamic allocation based on computational demands.

However, applying Expert-Race presents two challenges.

**Gating Function Conflict.** While softmax over the expert dimension is standard for score normalization in existing routing strategies, it disrupts cross-token score ordering in Race. Additionally, applying softmax across the full sequence incurs high computational costs and risks numerical underflow as sequence length grows. We therefore explore alternative activation functions, finding through Table 2 that the `identity` $\mathcal{G}(x) = x$ yields improved results.

**Training-Inference Mismatch.** Batch-wise candidate aggregation creates a fundamental mismatch between training and inference. During training, samples influence each other's routing selection and timesteps are randomly sampled per batch, whereas inference operates on independent samples with consistent timesteps. Since timesteps directly control noise mixing levels, this inconsistency degrades generation quality and can lead to model failure. At the same time, the mutual influence between samples during routing selection causes unstable inference. To mitigate these effects, we propose a learnable threshold $\tau$ that estimates the $\mathcal{K}$-th largest value through exponential moving average

**Algorithm 1** Pytorch-style Pseudocode of Expert-Race

```
# m: momentum
# tau: ema updated threshold
# x: input of shape (B, L, D)
# experts: list of FFN

# Compute router logits for each token
logits = router(x)    # (B, L, E)
score = logits.flatten()
gates = Identity()(logits) # activation
expect_k = B * L * k

# Get kthvalue and update threshold
if training:
    kth_val = kthvalue(score, k=expect_k)
    mask = score >= kth_val
    tau = m * tau + (1. - m) * kth_val
else:
    mask = score >= tau

# Aggregate the token for each expert
indices = dispatch(mask)
x = gather_input(x, indices)
gates = gather_gates(gates, indices)

# Process tokens by each expert and combine
outs = [experts[i](x[i]) for i in range(E)]
result = combine(outs, gates, indices)
```

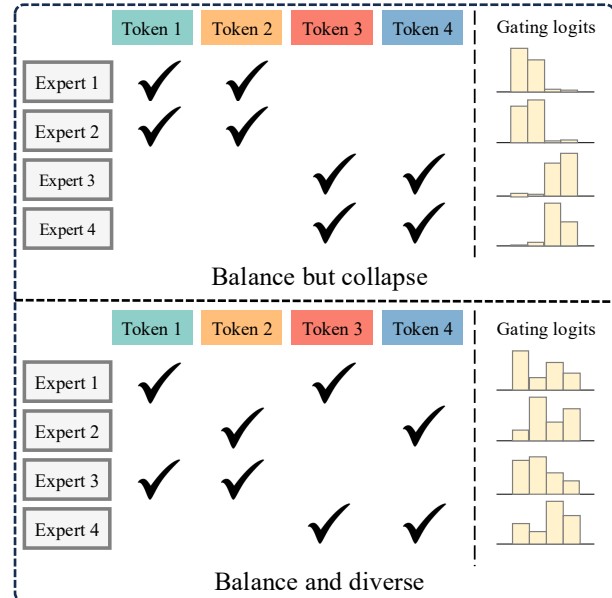

*Figure 4.* Toy examples of token assignment. Both of the two cases show perfect load balance that each expert process two tokens. But in the case above, experts 1 and 2 are assigned the same token, as are experts 3 and 4, where the 2-in-4 MoE collapse into 1-in-2. The example below shows a more diverse assignment, making full use of the expert specialization.

(EMA) updates during training.

$$\tau \leftarrow m\tau + (1 - m) \cdot \frac{1}{D_A} \sum_{i=1}^{D_A} \mathbf{S}'_{i,\mathcal{K}}, \quad (8)$$

where $\mathbf{S}'_{i,\mathcal{K}}$ represents the $\mathcal{K}$-th largest element in the $i$-th row of $\mathbf{S}'$. This adaptive threshold is directly applied during inference, ensuring sample independence and consistent performance.

**Pseudocode.** We provide core pseudocode in PyTorch style in Algorithm 1, illustrating how the expert selects the k-th largest logits and updates the threshold. Our algorithm is easy to implement, requiring only minor modifications to the existing MoE framework.

## 5. Load Balancing via Router Similarity Loss

In MoE systems, balanced token allocation across experts remains a critical engineering challenge. For our proposed Race strategy, the increased policy flexibility imposes greater demands on routing balancing.

**Mode Collapse in Balancing Loss.** The conventional balancing loss (Shazeer et al., 2017; Fedus et al., 2022), originally designed for token-choice, promotes load balance by enforcing uniform token distribution across experts, thereby preventing dominance by a small subset of experts. However, by only constraining the marginal distribution of scores

per expert, this approach fails to prevent collapse between experts with similar selection rules. As shown in Figure 4, if multiple experts follow the same rules for selecting tokens, they are downgraded to one wider expert. Although such configurations satisfy balance loss constraints, they undermine the specialization benefits of fine-grained expert design (Dai et al., 2024), ultimately degrading overall performance.

**Router Similarity Loss.** To tackle this issue, we propose maximizing expert specialization by promoting pairwise diversity among experts. Specifically, inspired by (Zbontar et al., 2021), we compute cross-correlation matrices and minimize their off-diagonal elements to encourage expert differentiation. Given the router logits $S \in \mathbb{R}^{(B \times L) \times E}$, we apply softmax along the expert dimension to obtain normalized probabilities $P$, and compute two correlation matrices

$$M' = M^T M, \quad P' = P^T P \quad (9)$$

where $M$ is the indicator matrix that $M_{i,j} = 1$ if expert $j$ selects the $i$-th token and 0 otherwise.

Then, we define the **router similarity loss**:

$$\mathcal{L}_{\text{sim}} = \frac{1}{T} \sum_{i,j \in [1,E]} W(i,j) \cdot P'_{i,j} \quad (10)$$

where $W(i,j)$ is the weighting function defined as:

$$W(i,j) = \begin{cases} \frac{M'_{i,j}}{\sum_{i=j} M'_{i,j}} \cdot E, & \text{if } i = j \\ \frac{M'_{i,j}}{\sum_{i \neq j} M'_{i,j}} \cdot (E^2 - E), & \text{if } i \neq j \end{cases} \quad (11)$$

In more detail, the off-diagonal elements denote the similarity between each pair of experts based on token selection patterns in the current batch. From a probabilistic perspective, $P'_{i,j}$ captures the joint probability of a token being routed to both expert $i$ and $j$. This formulation regularizes consistent co-selection patterns across experts while promoting diverse expert combinations. Regarding the diagonal elements, $P'_{i,i}$ represents a geometric mean version of the balance loss, effectively encouraging individual expert utilization.

## 6. Per-Layer Regularization for Efficient Training of Shallow Layers

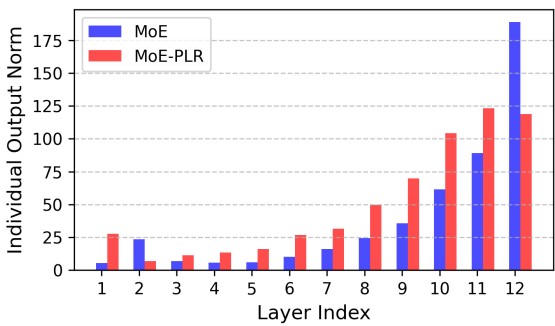

*Figure 5.* The norm of each block's output before added to the shortcuts. The output norm increases rapidly in deep layers, resulting in the weakening of shallow-layer components. This issue is alleviated with our proposed per-layer regularization.

DiT employs adaptive layer normalization (adaLN) when introducing conditions. In the pre-normalization (pre-norm) architecture, we observe that adaLN progressively amplifies the outputs of deeper layers. This causes the output magnitudes of shallow layers to be relatively diminished, as illustrated in Figure 5. This imbalance results in the learning speed of shallow layers lagging behind that of deeper layers, which is detrimental to the MoE training process. This imbalance has both advantages and disadvantages. On one hand, the outputs from deeper layers are more accurate, and their larger magnitudes make them less susceptible to the substantial noise present in shallow layers, facilitating more precise regression results. On the other hand, due to the presence of the normalization module in the final layer, the component of shallow layers in the residuals is diminished, posing a risk of gradient vanishing and resulting in the learning speed of shallow layers lagging behind that of deeper layers, which is detrimental to the MoE training process.

To mitigate this issue, we introduce a per-layer regularization that enhances gradients in a supervised manner without altering the core network structure. Specifically, given the hidden output $\mathbf{h}_l$ from the $l$-th layer, we add a projection layer $\mathcal{H} : \mathbb{R}^{L' \times d} \to \mathbb{R}^{L \times d}$ to predict the final target $\mathbf{y} \in \mathbb{R}^{L \times d}$, where $L$ and $L'$ represent the number of patches before and after the patchify operation. The per-layer loss is defined as:

$$\mathcal{L}_{\text{PLR}} = \mathbb{E}_{x_0,t,\epsilon,l} \left[ \frac{1}{N} \sum_{i=1}^{N} \left\| \mathbf{y}^{[n]} - \mathcal{H}(\mathbf{h}_l)^{[n]} \right\|^2 \right] \quad (12)$$

where $N$ is the total number of patches, and $n$ is the patch index. In our implementation, the projection layer is integrated into the MLP router as an additional prediction head. By supervising the projection layer's predictions against final targets, we enhance shallow layer contributions during training, improving overall MoE performance.

## 7. Experiments

**Implementation Details** We follow training configuration in (Peebles & Xie, 2023) and conduct experiments on ImageNet (Deng et al., 2009). The metrics used include FID (Heusel et al., 2017), CMMD (Jayasumana et al., 2024), and CLIP Score (Radford et al., 2021). We present a series of MoE size configurations, denoted as k-in-E where E represents the total number of experts and k indicates the number of average activated experts. Additionally, we set the inner hidden dimension of each expert to be $1/k$ of its dense counterpart to ensure that the number of activated parameters remains the same. For all experiments, we train DiT-B and its 2-in-8 MoE variant for 500K iterations unless specified otherwise.

**Routing Strategy** Expert-race enables extensive exploration within the logit space during training, enabling complexity-aware expert allocation across diffusion timesteps. As shown in Figure 6, our method dynamically assigns more experts to tokens at timesteps requiring higher image detail (lower timestep indices). In contrast, both token-choice and expert-choice strategies maintain fixed average expert allocations per timestep, lacking the temporal dynamic allocation capability. Within the framework proposed in Section 4.1, we conducted ablation studies on routing strategies for combinations of different dimensions. See Section 7 for more results.

**Gating Function** Table 2 shows that identity gating outperforms both softmax and sigmoid variants. In this experiment, we isolate other components (learnable threshold and regularizations) to verify the impact of the gating function on performance. We found that identity gating outperforms softmax and sigmoid under expert-race, and it enhances both token-choice and expert-choice compared to softmax.

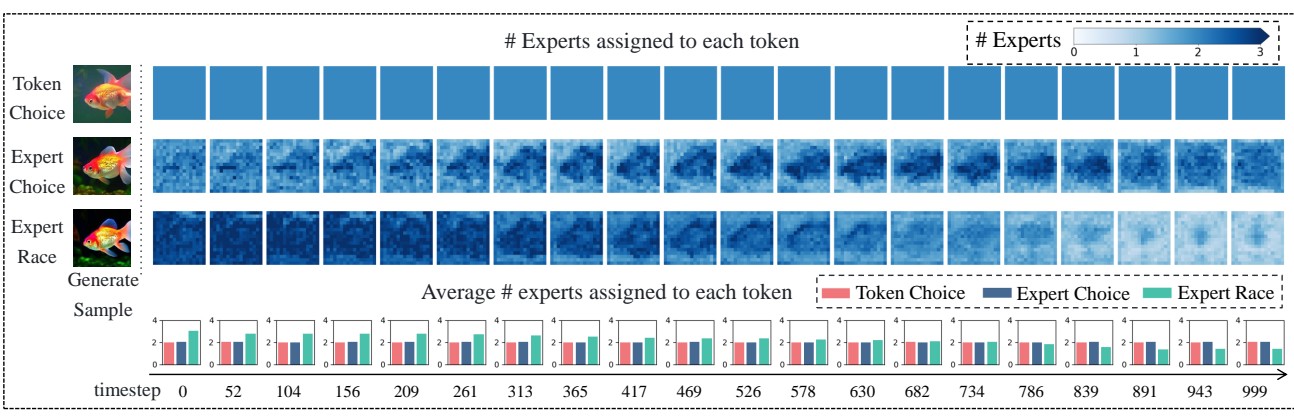

*Figure 6.* Average token allocation at different time steps. Expert-Race assigns more experts to the more complex denoising time steps, which occur at lower timestep indices that handle finer-grain image details.

*Table 2.* Routing strategy and gating function.

| Routing | Gating | FID↓ | CMMD↓ | CLIP↑ |
|---|---|---|---|---|
| Token Choice | | 17.28 | .7304 | 21.87 |
| Expert Choice | softmax | 16.71 | .7267 | 21.95 |
| Expert Race | | 16.47 | .7104 | 21.97 |
| Token Choice | | 15.25 | .6956 | 22.09 |
| Expert Choice | sigmoid | 15.73 | .6821 | 22.06 |
| Expert Race | | 13.85 | .6471 | 22.23 |
| Token Choice | | 15.98 | .6938 | 22.01 |
| Expert Choice | identity | 15.70 | .6963 | 22.04 |
| Expert Race | | **13.66** | **.6317** | **22.25** |

*Table 3.* Ablation study of core components.

| Setting | FID↓ | CMMD↓ | CLIP↑ |
|---|---|---|---|
| Expert Racing (softmax) | 16.47 | .7104 | 21.97 |
| + Identity Gating | 13.66 | .6317 | 22.25 |
| + Learnable Threshold | 11.56 | .5863 | 22.56 |
| + Per-Layer Reg. | 8.95 | .4847 | 22.94 |
| + Router Similarity | **8.03** | **.4587** | **23.09** |

*Table 4.* Load balance for 4-in-32 MoE.

| Setting | FID↓ | MaxVio↓ | Comb↑ |
|---|---|---|---|
| No Constraint | 11.38 | 6.383 | 18.98 |
| Balance Loss | 11.67 | 2.052 | 72.36 |
| Router Similarity | **10.77** | **0.850** | **83.10** |

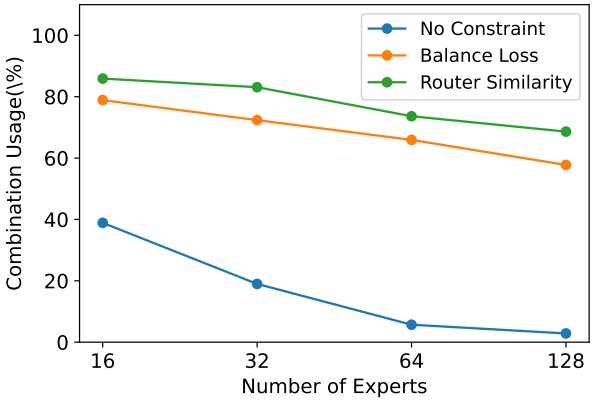

*Figure 7.* Combination usage between different number of experts.

**Load Balance** Table 4 compares our proposed router similarity loss with conventional load balancing loss (Dai et al., 2024). This setting is similar to the gating function ablation above, but here the MoE is 4-in-32 to further observe the impact of load balancing. The MaxVio (Wang et al., 2024b) metric measures how much the most violated expert exceeds its capacity limit. Combination Usage (Comb) measures the selection frequency of each expert pair across all tokens. It first sorts these frequencies in descending order, then counts the expert pairs that appear when the cumulative frequency reaches 95%, and finally ratios this count to the total number of possible combinations. This metric thus estimates the utilization of all expert pairs (higher is better).

Figure 7 further demonstrates the evaluation of MoE configurations across multiple scales (4-in-16,32,64,128), highlighting our approach's capability to diversify expert activation pattern compared to existing methods.

**Core Components** Table 3 demonstrates the improvements brought by each component. Starting from the baseline of expert racing with softmax gating, we incrementally added identity gating, learnable thresholds, per-layer regularization, and router similarity loss.

**Scaling Law** We first scale the base model sizes in the full pipeline, as detailed in Table 5. The comparison of our 4-in-32 MoE models with their Dense counterparts are across four sizes (B/M/L/XL). The number of activation parameter of Dense and MoE models are nearly identical. Results in Figure 1 demonstrate that our model significantly outperforms the corresponding Dense models given same activation parameter count. Furthermore, our MoE-4in32 model surpasses the XL-Dense model with less than half the total number of parameters, further showcasing the efficiency of our model design.

Table 5. Model specifications and evaluation results of the comparison between MoE and Dense models.

| Model Config. | Total Param. | Activate Param. | # Layers | Hidden | # Heads | FID↓ | CMMD↓ | CLIP↑ |
|---|---|---|---|---|---|---|---|---|
| B/2-Dense | 0.127B | 0.127B | 12 | 768 | 12 | 18.03 | .7532 | 21.83 |
| M/2-Dense | 0.265B | 0.265B | 16 | 960 | 16 | 11.18 | .5775 | 22.56 |
| L/2-Dense | 0.453B | 0.453B | 24 | 1024 | 16 | 7.88 | .4842 | 23.00 |
| XL/2-Dense | 0.669B | 0.669B | 28 | 1152 | 16 | 6.31 | .4338 | 23.27 |
| B/2-MoE-4in32 | 0.531B | 0.135B | 12 | 768 | 12 | 7.35 | .4460 | 23.15 |
| M/2-MoE-4in32 | 1.106B | 0.281B | 16 | 960 | 16 | 5.16 | .3507 | 23.50 |
| L/2-MoE-4in32 | 1.889B | 0.479B | 24 | 1024 | 16 | 4.04 | .2775 | 24.12 |
| XL/2-MoE-4in32 | 2.788B | 0.707B | 28 | 1152 | 16 | **3.31** | **.1784** | **24.68** |

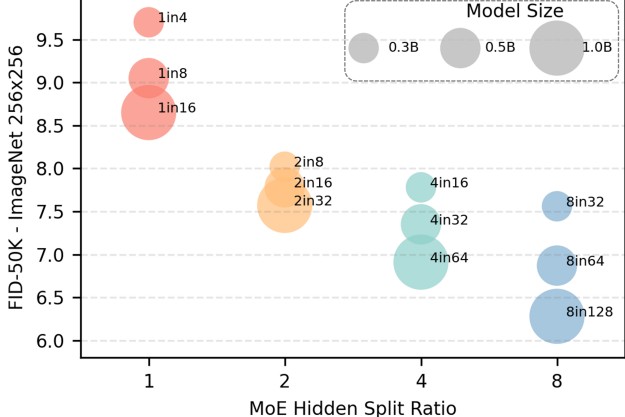

Figure 8. Scaling results of DiT-B in different MoE configurations. Our method demonstrates linear performance improvement when increasing expert split ratios and expanding candidate expert pools.

We further expand the model's size while maintaining the same number of active parameters under the same model configurations (DiT-B). The model expansion is achieved by varying the hidden split ratios of experts and increasing the number of candidate experts. As shown in Figure 8 and Table 6, increasing both the number of candidate experts and splitting the hidden dimensions of MoE leads to improvement in performance, highlighting the potential of our MoE architecture for scaling up.

**Extended Routing Strategies** We extend the token-choice and expert-choice routing strategies by introducing varying degrees of routing selection flexibility, aiming to investigate how training-stage router freedom impacts final model performance, as shown in Figure 9. All compared methods are trained for 500K iterations under identical configurations: a full pipeline with DiT-B/2-MoE-2-in-8 architecture. To ensure experimental consistency, all approaches employ learnable thresholds for inference queries. Specifically, we develop three new strategies: BL-Choice, BE-Choice, and LE-Choice, obtained through pairwise combinations of selection dimensions - Batch (B), Sequence Length (L), and Expert Count (E). Experimental results in Table 7 demonstrate that strategies with higher train-

Table 6. Evaluation results of different MoE configurations with the same number of activation parameters.

| MoE Config. | Hidden Split | Total Param. | FID↓ | CMMD↓ | CLIP↑ |
|---|---|---|---|---|---|
| 1-in-4 | | 0.297B | 9.70 | .5200 | 22.82 |
| 1-in-8 | 1 | 0.524B | 9.05 | .4976 | 22.91 |
| 1-in-16 | | 0.978B | 8.65 | .5019 | 22.92 |
| 2-in-8 | | 0.297B | 8.03 | .4587 | 23.09 |
| 2-in-16 | 2 | 0.524B | 7.78 | .4607 | 23.06 |
| 2-in-32 | | 0.977B | 7.57 | .4483 | 23.12 |
| 4-in-16 | | 0.297B | 7.78 | .4628 | 23.09 |
| 4-in-32 | 4 | 0.524B | 7.35 | .4460 | 23.15 |
| 4-in-64 | | 0.977B | 6.91 | .4244 | 23.21 |
| 8-in-32 | | 0.297B | 7.56 | .4516 | 23.11 |
| 8-in-64 | 8 | 0.524B | 6.87 | .4263 | 23.24 |
| 8-in-128 | | 0.977B | 6.28 | .4015 | 23.35 |

ing selection freedom generally outperform conventional fixed-dimension top-k selection approaches (such as token-choice/expert-choice). Notably, Expert Race achieves the best performance across all evaluated routing strategies.

**Comparison with Leading Methods** We further provide a comparison with leading approaches shown in Table 8, where *Samples* is reported as *training iterations × batch size*. The classifier-free guidance scale is 1.5 for evaluation.

We train a vanilla DiT (marked with *) and a MoE model with Expert Race following the training protocol from the original DiT paper (Peebles & Xie, 2023), except that we use a larger batch size 1024 to improve memory utilization and train for only 1.75M steps. The learning rate remains unchanged at $10^{-4}$. Our MoE model achieves better performance with a similar activated parameter amount,

**Computational Overhead** During training, Expert-Race maintains the same total number of tokens assigned to experts as other routing methods, differing only in their distribution. The load imbalance can be mitigated through router similarity regularization, thus the training efficiency of Expert-Race is comparable to other routing methods. However, during inference, samples within a batch share identical timesteps. Variations in expert activations across different timesteps cause fluctuations in computational cost.

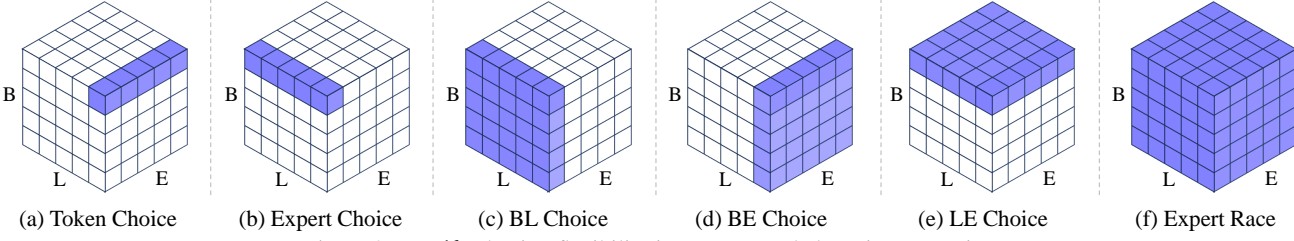

*Figure 9.* Top-$\mathcal{K}$ selection flexibility in more extended routing strategies.

*Table 7.* Design Choices and Evaluation Results of Different Routing Strategies

|  | Token Choice | Expert Choice | BL Choice | BE Choice | LE Choice | Expert Race |
|---|---|---|---|---|---|---|
| $D_A$ | $B*L$ | $B*E$ | $E$ | $L$ | $B$ | $1$ |
| $D_B$ | $E$ | $L$ | $B*L$ | $B*E$ | $L*E$ | $B*L*E$ |
| $\mathcal{K}$ | $k$ | $k*L/E$ | $B*L*k/E$ | $B*k$ | $L*k$ | $B*L*k$ |
| FID↓ | 9.50 | 10.13 | 9.08 | 8.28 | 8.89 | **8.03** |
| CMMD↓ | .5202 | .5639 | .5145 | .4636 | .4871 | **.4587** |
| CLIP↑ | 22.81 | 22.73 | 22.87 | 23.05 | 22.99 | **23.09** |

*Table 8.* Comparison with other methods on ImageNet 256x256.

| Model Config. | Total | Activated | Samples | FID↓ | IS↑ | Precision↑ | Recall↑ |
|---|---|---|---|---|---|---|---|
| ADM-G (Dhariwal & Nichol, 2021) | 0.608B | 0.608B | 2.0M × 256 | 4.59 | 186.70 | 0.82 | 0.52 |
| LDM-8-G (Rombach et al., 2022a) | 0.506B | 0.506B | 4.8M × 64 | 7.76 | 209.52 | 0.84 | 0.35 |
| MDT (Gao et al., 2023) | 0.675B | 0.675B | 2.5M × 256 | 2.15 | 249.27 | 0.82 | 0.58 |
| MDT (Gao et al., 2023) | 0.675B | 0.675B | 6.5M × 256 | 1.79 | 283.01 | 0.81 | 0.61 |
| DiT-XL/2 (Peebles & Xie, 2023) | 0.669B | 0.669B | 7.0M × 256 | 2.27 | 278.24 | 0.83 | 0.57 |
| SiT-XL (Ma et al., 2024) | 0.669B | 0.669B | 7.0M × 256 | 2.06 | 277.50 | 0.83 | 0.59 |
| MaskDiT (Zheng et al., 2023) | 0.737B | 0.737B | 2.0M × 1024 | 2.50 | 256.27 | 0.83 | 0.56 |
| DiT-MoE-XL/2 (Fei et al., 2024) | 4.105B | 1.530B | 7.0M × 1024 | 1.72 | 315.73 | 0.83 | 0.64 |
| DiT-XL/2* | 0.669B | 0.669B | 1.7M × 1024 | 3.02 | 261.49 | 0.81 | 0.51 |
| RaceDiT-XL/2-4in32 | 2.788B | 0.707B | 1.7M × 1024 | 2.06 | 318.64 | 0.83 | 0.60 |

*Table 9.* Computational Overhead. Inference evaluation of the number of activated experts per token on RaceDiT-XL/2-4in32. The first column refers to batch size and * denotes the batch is composed of samples from different timesteps.

|  | Mean | | Max | |
|---|---|---|---|---|
| Batch Size | t=0 | t=1000 | t=0 | t=1000 |
| 1 | 5.35 | 3.79 | 5.81 | 4.56 |
| 32 | 5.40 | 3.78 | 5.48 | 4.01 |
| 32* | 4.03 | | 4.27 | |

As shown in Table 9, the peak computational cost reaches 5.81 experts per token, potentially limiting inference efficiency. To address this issue, one solution is to construct batches containing samples from multiple timesteps. This can be achieved through asynchronously interleaving samples from different timesteps within one batch, similar to pipeline parallelism. Such an approach can reduce the peak cost to 4.27 experts per token and the average cost to 4.03, significantly mitigating dynamic routing overhead.

## 8. Conclusion

This paper proposes Expert Race, a novel Mixture-of-Experts (MoE) routing strategy that enables stable and efficient scaling of diffusion transformers. Compared to previous methods with fixed degrees of freedom in expert-token assignments, our strategy achieves higher routing flexibility by enabling top-k selection across the full routing space spanning batch, sequence, and expert dimensions. This expanded selection capability provides greater optimization freedom, significantly improving performance when scaling diffusion transformers. To address challenges from increased flexibility, we propose an EMA-based threshold adaptation mechanism that mitigates timestep-induced distribution shifts between training (randomized per-sample timesteps) and inference (uniform timesteps), ensuring generation consistency. Additionally, per-layer regularization improves training stability, while router similarity loss promotes diverse expert combinations and better load balancing, as shown on 256×256 ImageNet generation tasks. As a general routing strategy, future work will extend Expert Race to broader diffusion-based visual tasks.

## Impact Statement

We consistently adhere to ethical standards and responsible AI development principles. Our models are trained on the widely-used, open-source ImageNet dataset, ensuring diverse and representative outputs while minimizing the risk of biased or unsuitable content. We are committed to rigorously reviewing the content generated by our models to prevent any harmful material.

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
