# OpenReview forum: "Expert Race: A Flexible Routing Strategy for Scaling Diffusion Transformer with Mixture of Experts"
_ICML.cc/2025/Conference — ICML 2025 poster_

### Official Review · Reviewer_ERH6 · 2025-03-09

**Overall Recommendation:** 3

**Summary:**

The authors introduce a novel Mixture-of-Expert model for Diffusion Transformers. The main novelty aspect lies in its routing strategy, Expert Race, which allows to tune experts assignment not only by token, but also by batch element (ie, by time-step in the diffusion process). As a consequence, the model gains the additional flexibility of employing a different number of experts depending on the stage of the diffusion process it finds itself: arguably, this is beneficial, as at the beginning of the denoising procedure (when the image is still mostly noise) noise identification is simpler than towards the end (where the denoising procedure is almost complete, and more fine-grained details must be generated). While granting additional flexibility, this strategy comes with its challenges to ensure effective training. To address these, the paper proposes a number of adjustments to the training procedure, mainly the Router Similarity loss, to ensure experts specialisation, and a per-layer regularisation, to boost gradient in shallower model layers. Additional modifications ensure stability (dropping softmax as gating function) and consistency in predictions between training and inference (this is necessary due to the different distributions of time-steps in a batch seen by the model during these two phases).
The effectiveness of the model is tested by training on ImageNet and reporting FID, CMMD, and CLIP scores of the distribution of generated images, comparing them against alternative routing strategies available in the literature. Ablation studies confirm the advantages provided by the various adaptations introduced to the training procedure. Scaling laws confirm that the improvements are sustained also for large model sizes (up to ~O(1B)).

**Claims And Evidence:**

Given the results provided, I’m reasonably convinced of the validity of the method.

**Essential References Not Discussed:**

The references provided are in my opinion sufficient to properly contextualise this work.

**Experimental Designs Or Analyses:**

Code wasn’t made available, so I couldn’t skim through their implementation. From what I could read, the experiments design seems solid.

**Methods And Evaluation Criteria:**

Evaluation criteria are based on commonly used metrics for image quality evaluation (FID, CMMD, CLIP), and training is done on the ImageNet dataset. Both these choices make perfect sense to me.

**Other Comments Or Suggestions:**

- L63,C1 “flexibilities greatly boost” -> “flexibility greatly boosts”
- L111,C1 “share” -> “shares”
- L134,C2 “build a set with \mathcal{K} largest value in tensor” -> “builds a set with the \mathcal{K} largest value in a tensor”
- L281,C2: “patchily operation ()” -> reference missing?

**Other Strengths And Weaknesses:**

The paper is -for the most part- explained well, with the main idea illustrated clearly, and its element of novelty wrt previous work properly defined. The structure is solid, and the story reads well. Results are reasonably detailed and convincing.

**Questions For Authors:**

- Q1 On training vs inference - I’m honestly having a hard time understanding how your method works at inference/generation. If I got it correctly, at training time the routing strategy is deciding how to allocate the various experts while looking at the *whole* batch, which includes various samples at different time-steps / noise levels. This way, the model can learn to adaptively allocate more experts to less noisy tokens, if needed. Now, at generation, the time-steps distribution within the batch (if you’re doing batch generation at all!) is clearly different: first we de-noise samples that are all at time $t=T$, then all at $t=T-1$, and so on until $t=0$. I reckon this is the mismatch you hint at in the Training-Inference Mismatch paragraph. What I don’t understand is why the choice $\tau$ in Alg1 should help in this regard? I was expecting that some sort of explicit (learnable) dependency on $t$ was gonna be baked into the threshold directly?
- Q2 Considerations on workload during generation - From my understanding (correct me if I’m wrong), both Token-choice and Expert-choice guarantee constant workload per time-step at generation, in the sense that, for each $t$, there is a certain fixed number of $E_i(X_j)$ functions that must be evaluated (although the distributions of tokens per expert / expert per token might vary). In your case, this is not valid anymore (Fig6 highlights this). I was wondering what repercussions this had, if any, on the total wall-clock time / memory resources necessary for generation: it’s hard for the gauge this from the top of my head, as I reckon it depends on code implementation and parallelisability considerations. To help clarify what I mean, consider the extreme case where only 1 expert is invoked for each time-step between $T$ and 1, and all remaining $k-1$ expert invocations are used at the very last diffusion step $t=0$. If all experts invocations are perfectly parallel and we have infinite resources, at the end of the day the overall time for generation doesn’t change, but the last step would generally require a spike in computational demands, which might bottleneck operations and slow down generation. Does this make sense? If so, could you comment of this?

**Relation To Broader Scientific Literature:**

The method proposed in the paper is grounded in a direct generalisation of MoE routing strategy for diffusion models, expanding both (Zoph et al., 2022, Zhou et al., 2022)

**Theoretical Claims:**

The only theoretical claim casts the proposed Router Similarity loss to a generalised load balance loss. I haven't checked the derivation in detail, but seems legit at a glance.

---

### Official Review · Reviewer_h4on · 2025-03-14

**Overall Recommendation:** 3

**Summary:**

- This paper proposes an innovative MoE block integrated with a simple routing strategy and multiple regularization techniques within diffusion transformer frameworks, to concurrently optimize performance and computational efficiency. In particular, the authors broaden the routing strategy's exploration space by simultaneously considering token, batch, and expert dimensions, effectively addressing limitations inherent in existing methods that primarily emphasize either expert or token dimensions alone. Moreover, grounded in thorough empirical observations and logical justification, the authors substitute the conventional softmax routing mechanism with an identity function, complemented by a learnable threshold designed to alleviate inconsistencies arising from noise magnitude discrepancies between training and inference phases. Drawing inspiration from the Barlow Twins framework, the authors further propose an analogous loss formulation to combat mode collapse. Additionally, an auxiliary regularization branch is introduced to maintain stable output magnitudes across layers, thereby preventing gradient vanishing issues during model training. Empirical evaluations demonstrate that the proposed approach yields substantial performance gains on image generation benchmarks using the ImageNet dataset, thereby validating its effectiveness and advancing the state-of-the-art in this domain.

**Claims And Evidence:**

- In general, the claims of this paper are supported by clear and convincing evidence.

**Essential References Not Discussed:**

- In general, this paper discusses related approaches adequately.

**Experimental Designs Or Analyses:**

- Yes, I have assessed the soundness and validity of the experimental designs and analyses presented in the paper. Regarding the experimental designs, comparisons with similar approaches are limited; however, this limitation arises primarily due to the scarcity of existing methods related to diffusion models, particularly those employing DiT. Also, the analysis is adequate considering the contents in the supplementary material.

**Methods And Evaluation Criteria:**

- Evaluating the proposed methods solely on ImageNet for image generation may not be sufficient. Even when focusing exclusively on image generation (without considering other modalities), additional datasets such as COCO could offer valuable insights. Furthermore, FID is no longer considered the most suitable metric for open-domain image generation; alternative metrics like ImageReward or HPS might provide more meaningful evaluations.

**Other Comments Or Suggestions:**

- The figures are somewhat low in quality; I suggest that the authors use vector graphics for better clarity.

**Other Strengths And Weaknesses:**

- This paper is well-crafted and effectively articulates both the proposed methodology and the corresponding experimental outcomes.
- The implementation of the approach is methodical and straightforward, facilitating its practical applicability.
- Comprehensive implementation details significantly enhance the reproducibility of the research.
- However, certain aspects lack sufficient discussion, such as the influence of timestep and effects at different resolutions.

**Questions For Authors:**

- (1) From my perspective, the influence of timestep selection on the MoE, as well as the impact of varying input complexities (e.g., performance differences across categories), is particularly intriguing. Although the authors provide a basic illustration and a brief subsection discussing these aspects, incorporating theoretical analyses would further enhance the contribution of this paper.
- (2) I'm curious about the consistency and generalizability of the proposed method across different resolutions or datasets, such as COCO.
- (3) Additionally, what performance does the method achieve when evaluated using alternative metrics like ImageReward or HPS?

**Relation To Broader Scientific Literature:**

- From my perspective, the key contribution of this paper is introducing a simple yet efficient MoE to the DiT architecture, accompanied by theoretical analysis. Given that very few related approaches exist, this paper makes a meaningful contribution to the broader scientific literature.

**Theoretical Claims:**

- I have reviewed the theoretical claims presented in Section C of the supplementary material and found them to be correct.

---

### Official Review · Reviewer_qFyg · 2025-03-16

**Overall Recommendation:** 4

**Summary:**

This paper introduces Race-DiT, a novel approach for applying Mixture of Experts (MoE) to diffusion transformers with a flexible routing strategy called Expert Race. The key innovation is allowing tokens and experts to compete together in a global selection process, enabling dynamic allocation of computational resources based on token complexity. The authors also propose per-layer regularization to address shallow layer learning challenges and router similarity loss to prevent mode collapse among experts. Experiments on ImageNet demonstrate significant performance gains over dense models and other MoE routing strategies. The method shows promising scaling properties with both increased expert numbers and hidden dimension splits. Overall, Race-DiT achieves better FID scores with fewer activated parameters compared to traditional dense models, showing the potential of flexible MoE routing in diffusion models.

**Claims And Evidence:**

Per-layer Regularization lacks strong theoretical evidence. While the authors identify that shallow layers contribute less to DiT's pre-norm architecture, the fundamental necessity for layer balance is inadequately justified. The ablation study shows empirical improvements but doesn't conclusively demonstrate why balancing shallow and deep layers is optimal versus simply allowing deeper layers to dominate. The visual evidence in Figure 5 shows the phenomenon clearly, but the causal relationship between layer balance and overall performance could use more rigorous analysis. Additionally, the interaction between the MoE architecture and layer dynamics needs more theoretical grounding beyond empirical results.

**Essential References Not Discussed:**

The paper lacks a reference to "Addressing negative transfer in diffusion models" (Go et al., NeurIPS 2023), which provides explicit evidence for diffusion's multi-task nature through demonstrated negative transfer effects across timesteps. Race-DiT repeatedly emphasizes diffusion's varying temporal complexity as motivation for flexible routing, and Go et al.'s work would significantly strengthen this foundational claim with empirical support.

**Experimental Designs Or Analyses:**

The experimental design is generally valid, but comparisons with other MoE-based diffusion research are somewhat limited. The internal comparisons between different MoE strategies (Expert Race vs. Token/Expert Choice) are comprehensive.

However, the paper fails to clearly quantify the additional computational complexity and actual computational cost of Expert Race compared to Expert Choice or Token Choice. Since Expert Race requires flattening and processing the entire score tensor, it likely has poorer cache efficiency and higher computational overhead. This trade-off between quality improvement and computational cost is not explicitly analyzed, which is particularly concerning for an optimization technique like MoE where efficiency is a primary consideration.

**Methods And Evaluation Criteria:**

The evaluation methodology generally makes sense for diffusion models.

However, the computational efficiency comparison between different MoE strategies (Expert Race vs. Token/Expert Choice) is notably absent. Given that MoE's primary advantage is computational efficiency, understanding the overhead introduced by global token selection would be crucial. The paper focuses heavily on quality metrics (FID, CMMD, CLIP) but lacks throughput, latency, or memory usage comparisons. Additionally, while they show strong performance against dense models, evaluation against other contemporary diffusion MoE approaches like EC-DiT or Raphael is limited to the appendix rather than being central to the main evaluation.

**Other Comments Or Suggestions:**

The paper would benefit from a brief discussion of inference-time optimizations for Expert Race, since the global top-k selection might introduce latency challenges in production environments. Consider adding visualization of what different experts specialize in, perhaps showing attention maps or feature visualizations from different experts. Figure 6 currently shows allocation patterns but doesn't reveal what features drive those allocations. A minor typo appears in equation (7) where the summation indices could be clarified.

**Other Strengths And Weaknesses:**

# Strengths

- The model achieves impressively lower FID scores compared to dense models even when considering total parameter count, not just activated parameters, demonstrating true parameter efficiency.
- The proposed Expert Race mechanism effectively addresses temporal complexity variation in diffusion models, showing marked improvement in early denoising timesteps where detail preservation is critical.
- The authors perform extensive ablation studies that clearly isolate the contribution of each proposed component.
- Router Similarity Loss shows strong theoretical grounding with clear connections to existing balance loss formulations.

# Weaknesses

- The dramatic change in layer dynamics after applying Per-Layer Regularization, particularly in layers 1-2, lacks detailed analysis of why these specific layers are most affected.
- The paper provides limited analysis of expert specialization patterns - what specific features or timestep characteristics different experts learn to handle.
- Testing is limited to ImageNet at 256×256 resolution without validation on more diverse datasets or higher resolutions.
- The throughput-vs-quality tradeoff analysis is insufficient for practical deployment considerations.

**Questions For Authors:**

- How does Expert Race compare to Token/Expert Choice in terms of computational overhead during both training and inference? Could you provide wall-clock time comparisons on equivalent hardware?
- Have you explored whether the benefits of Per-Layer Regularization are specific to MoE architectures, or would dense DiT models also benefit from this approach?

**Relation To Broader Scientific Literature:**

The paper makes a significant contribution by expanding MoE routing to consider all dimensions (B, L, E) with equal weight. Particularly valuable is connecting this approach to diffusion models' unique temporal characteristics across different denoising timesteps. By recognizing that diffusion complexity varies both spatially (across image regions) and temporally (across timesteps), the authors provide insights applicable beyond this specific implementation. This connection between model architecture design and diffusion's inherent multi-task nature advances the field's understanding of how to build more efficient generation models.

**Theoretical Claims:**

I checked the correctness of the Analysis of Router Similarity Loss section. The derivation showing how router similarity loss extends traditional balance loss by considering pairwise expert interactions is mathematically sound.

---

### Decision · Program_Chairs · 2025-05-01

**Decision:**

Accept (poster)

**Comment:**

This work proposes a new technique to implement MoE that allows expert selection by considering both spatial image regions and temporal denoising steps. This allows for more flexibility and cost reduction. Moreover, the authors propose a mode collapse balancing loss and per-layer regularization to prevent vanishing gradients.

The experimental results seem to convince the reviewers in general.

The rebuttal from the authors was not delivered to the reviewers since it was submitted as AC private message slightly after the deadline. Therefore, there has been no modification to the original scores during the rebuttal period.